# Physicochemical Characterization of Packaging Foils Coated by Chitosan and Polyphenols Colloidal Formulations

**DOI:** 10.3390/ijms21020495

**Published:** 2020-01-13

**Authors:** Lidija Fras Zemljič, Olivija Plohl, Alenka Vesel, Thomas Luxbacher, Sanja Potrč

**Affiliations:** 1Laboratory for Characterization and Processing of Polymers, Faculty of Mechanical Engineering, University of Maribor, Smetanova 17, SI-2000 Maribor, Slovenia; olivija.plohl@um.si (O.P.); sanja.potrc@um.si (S.P.); 2Department of Surface Engineering and Optoelectronics, Jožef Stefan Institute, Teslova 30, SI-1000 Ljubljana, Slovenia; alenka.vesel@ijs.si; 3Anton Paar GmbH, Anton-Paar-Street 20, A-8054 Graz, Austria; thomas.luxbacher@anton-paar.com; 4Faculty of Chemistry and Chemical Engineering, University of Maribor, Smetanova 17, SI-2000 Maribor, Slovenia

**Keywords:** active food packaging, chitosan, extracts, UV/ozone (UV/O_3_), PE/PP functionalization, surface properties

## Abstract

In this research, antimicrobial polysaccharide chitosan was used as a surface coating for packaging material. The aim of our research was to establish an additive formulation of chitosan and antioxidative plant extracts as dispersion of nanoparticles. Chitosan nanoparticles with embedded thyme, rosemary and cinnamon extracts were synthesized, and characterized for this purpose. Two representative, commercially used foils, polypropylene (PP) and polyethylene (PE), previously activated by UV/ozone to improve coating adhesion, were functionalized using chitosan-extracts nanoparticle dispersions. The foils were coated by two layers. A solution of macromolecular chitosan was applied onto foils as a first layer, followed by the deposition of various extracts embedded into chitosan nanoparticles that were attached as an upper layer. Since active packaging must assure bioactive efficiency at the interface with food, it is extremely important to understand the surface characteristics and phenomena of functionalized foils. The physico-chemical analyses of functionalized foils were thus comprised of surface elemental composition, surface charge, wettability, as well as surface morphology. It has been shown that coatings were applied successfully with an elemental composition, surface charge and morphology that should enable coating stability, homogeneity and consequently provide an active concept of the packaging surface in contact with food. Moreover, the wettability of foils was improved in order to minimize the anti-fogging behavior.

## 1. Introduction

Packaging continues to be one of the most important and innovative areas for the development of new processes and products. Besides providing protection, specific physico-chemical properties and an attractive esthetical look, these products function as a preservation system. Nowadays, food preservation, quality maintenance and safety are major growing concerns of the food industry. In the past years it is evident that, over time, consumers’ demands for natural and safe food products with stringent regulations to prevent food-borne infectious diseases has increased [1,2]. Thus, the trend of developing new packaging is the directive to develop intelligent, biodegradable, active, human and environmentally friendly packaging, using mostly: vacuum treatment, sterilization, freshness and time–temperature indicators, oxygen and leak indicators, color measurement sensors, ethanol emitters, ethylene absorbers, etc., and specific active agents providing antimicrobial and antioxidant functionalities [3,4,5,6,7]. Active packaging is able to modify the condition of the packed food without provoking any substantial variation in its quality and nutritional value, while improving its shelf life and, ultimately, its safety. The use of antimicrobials that inhibit or inactivate microorganisms can reduce foodborne diseases significantly. Globally, approximately 2.2 million people are killed annually because of foodborne and waterborne diseases, with numbers continuing to rise due to an increase in the resistance of pathogens and the emergence of new pathogens. Moreover, through the extension of the shelf life time, active packaging offers to food manufacturers the prolongation of food distribution distances, and, thus, opening the export of products to new markets around the world, which is as an essential tool for market growth [7].

Various antimicrobial agents may be incorporated [8] in the packaging or food system, which are chemical antimicrobials, antioxidants, biotechnological products such as enzymes, antimicrobial polymers, natural antimicrobials, such as extracts and gas. Antimicrobial agents can be classified into three groups: chemical agents, natural agents, and probiotics [9]. Due to an increased health and environmental awareness, there is a growing demand for natural substances to be applied as activators in the development of active packaging. Among natural substances, biopolymers such as polysaccharides are very attractive. It has been shown that films and coatings of chitosan and poly-L-lysine applied to the food package possess inherently antimicrobial properties through inhibiting the microbial growth by causing leakage of intracellular constituents of microbial cells [10,11]. Unfortunately, it has been found that most of the polysaccharides show very poor antioxidant activity [3,12], which is an extremely welcomed property of bioactive packaging materials to retard the natural processes (i.e., the oxidative deterioration of food products such as meat is caused by the degradation reactions of fats and pigments [13]), causing food spoilage by reducing oxygen and moisture. Since essential oils are rich in volatile terpenoids and phenolic particles, they have the potential to inhibit oxidation processes, as well as a wide spectrum of microorganisms [14]. Several researchers combine those essential oils with biopolymers, in order to widen the activity spectrum of antimicrobial agents, increase the mode of action (antioxidant and antimicrobial activity) and improve the condition of targeted microorganisms’ inhibition. Although many strategies to combine polysaccharides and essential oils, as well as many coating technologies, exist [15,16,17,18,19,20,21,22,23], there are still many challenging opportunities for the combination of natural compounds as a packaging coating. With this fact in our minds, we developed a bilayer coating for the most representative synthetic packaging materials, i.e., foils of polyethylene (PE) and polypropylene (PP). Prior to coating, the polymer foils were activated by UV/ozone (UV/O_3_), in order to increase their surface free energy and to improve the binding capacity, as well as the stability of the coatings. The first layer consists of macromolecular chitosan solution and facilitates excellent antibacterial properties, while the second (uppermost) layer contains a nano - dispersed network of polyphenol extracts embedded into chitosan nanoparticles, which should enable antioxidant and antimicrobial properties simultaneously. With the application of this bilayer strategy, barrier properties may also be significantly improved. At the contact points of the active packaging materials’ surface with food it is extremely important to assure bioactive efficiency, as well as safety, which is very much dependent on the adsorption and desorption phenomena at the interfaces by creation of active packaging using surface activators. Thus, it is extremely important to characterize the surface properties of functionalized packaging materials that will be further subject to contact with specific foods. Among the surface parameters, surface charge is a central parameter for the enhancement or suppression of the interaction between dissolved compounds in an aqueous solution and solid material surfaces. The zeta potential, as the indicator for solid surface charge, is a valuable parameter for the comparison of material surfaces before and after surface treatment, as well as for their charging behavior in an aqueous solution. In recent years, increased interest in the determination of the electrokinetic or zeta potential of macroscopic solid surfaces of fibrous substrates and flat sheets has led to the establishment of the streaming potential method [24].

This paper covers the application of the streaming potential method to monitor the charging behavior and electrokinetic response of PP and PE foils functionalized by macromolecular chitosan solutions as the first coating layer, and chitosan nanoparticles with embedded polyphenols/extracts (thyme, rosemary and cinnamon extract) as the second layer. This is especially important, in order to follow the protonation behavior of chitosan amino groups and the dissociation of phenolic groups, which create the antimicrobial and antioxidant activity at the foil surface. It is known that protonated amino groups are responsible for the antimicrobial activity of chitosan [25,26], whilst the dissociation degree of phenolic groups influences the antioxidant activity [27]. Considering the final application of the chitosan–polyphenol functionalized PP and PE foils as packaging materials, it is of paramount importance to understand their surface properties better, and, consequently, their response in a given environment. The detailed physico-chemical characterization of packaging foils with a novel functionalization by the synergistic colloidal formulation of chitosan and different polyphenols is presented, whilst the resulting bioactive character of these foils will be discussed in a subsequent paper; hitherto, to the best of the authors’ knowledge no such detailed study has been presented previously.

## 2. Results and Discussion

### 2.1. Dispersion Characterization

#### 2.1.1. Particle Size, PDI and Zeta Potential Determination

The procedure used for chitosan nanoparticles is ionic gelation, which is known from the literature to provide nano-sized particles. The size (hydrodynamic diameter), polydispersity index (PDI) and zeta potential (ZP) of synthesized nanoparticles determined by dynamic and electrophoretic light scattering, DLS and ELS, are given in Table 1 for dispersions of chitosan nanoparticles (CSNP) and chitosan nanoparticles with embedded thyme extract (CSNP THY), rosemary extract (CSNP ROS), and cinnamon extract (CSNP CIN). The results are presented as the average of three measurements. 

It is seen clearly that the incorporation of polyphenols into chitosan nanoparticles increased the hydrodynamic diameter, which is the proof for a successful embedding. The increase in the size of chitosan nanoparticles with an incorporated active substance (polyphenols in extracts) from the nano (colloidal) to the micro scale, confirms the binding of the active ingredients to the interior, or onto the surface of the chitosan nanoparticles. The increase in size was the most pronounced for the rosemary extract. It seems that this extract was embedded into chitosan nanoparticles in a way that there was a little heterogeneity from a particles’ size (PDI 0.642) point of view. For CSNP with thyme or cinnamon extracts the PDI was higher, which indicates a higher heterogeneous size distribution of nanoparticles. At pH 4 the zeta potential of all dispersions was higher than +30 mV, showing dispersion stability due to sufficiently high repulsive forces between nanoparticles. The positive zeta potential further confirms the accessibility of amino groups of chitosan, which suggests the encapsulation of the polyphenols into the chitosan nanoparticles. 

It has to be noted that the nanoparticles expression is meant for basic chitosan nanoparticles, which of course, were larger after encapsulation of polyphenols. Basic chitosan nanoparticles were of size: 293 ± 80 nm. Because measurement provides hydrodynamic radio, pure particles (without water molecules surrounded) are smaller and thus to our opinion in nano-scale which was also proved by SEM (see below). 

#### 2.1.2. Encapsulation Efficiency

The encapsulation efficiency, or degree of encapsulation (%) of thyme extract, rosemary extract and cinnamon extract in chitosan nanoparticles was determined using UV spectroscopy. Using Equation (1), the degree of encapsulation was calculated, and values are presented in Table 2. The results are given as an average of three measurements.

The obtained results show a very high (more than 75%) and similar encapsulation degree for all extracts embedded in chitosan nanoparticles. However, in our approach, coatings existed from nanoparticles dispersed in the rest of the chitosan and extracts’ solution, with the strategy that an increase in stability could be achieved by simply incorporating the nanoparticles into another macromolecular layer. The improved stability of attached nanoparticles was, besides this strategy, also guaranteed by the first coated layer of foils using a 2% chitosan macromolecular solution.

#### 2.1.3. Total Phenolic Content

The total phenolic contents of thyme, rosemary, and cinnamon, as well as of chitosan nanoparticles with embedded extracts, are shown in Table 3.

The total phenolic content is expressed as mg GA/g of extract (*W*_GA_). The concentration of GA was highest in the cinnamon extract solution, followed by the rosemary extract solution. The presence of GA was significantly lower in formulations with CSNP, which again confirms that the extracts were encapsulated in the chitosan nanoparticles, and, due to their poor solubility, kept the extracts embedded. Furthermore, chitosan did not show any antioxidant behavior and, thus, did not contribute to the determination of the phenolic content. The higher total phenolic content was observed for CSNP CIN, which correlates with the result for pure CIN, which was the richest in phenols.

### 2.2. Functional Foils

#### 2.2.1. ATR-FTIR Spectroscopy

The different functionalization steps of the phenolic-embedded NPs’ deposition onto different foils was followed by Attenuated Total Reflection–Fourier Transform Infra-Red (ATR-FTIR) spectroscopy (Figure 1). Typical peaks for unmodified PE and PP foils are shown as reference spectra on all IR graphs. The bands at 2911 cm^−1^ can be assigned to CH_2_ asymmetric stretching, the band at 2849 cm^−1^ to CH_2_ symmetric stretching, while the bands at 1463 and 718 cm^−1^ correspond to bending and rocking deformation. Some typical bands for PP are the same as for PE, otherwise at 2951 cm^−1^ CH_3_ stretching, 2862 cm^−1^ CH_2_ asymmetric stretching, and CH_2_ and CH_3_ bending vibrations were observed at 1461 cm^−1^ and 1374 cm^−1^. Bands assigned to CH_3_ and CH_2_ rocking vibrations for PP are presented at lower wavenumbers and in lower intensities, i.e., at 997 cm^−1^ and 973 cm^−1^. The spectrum (labeled as red in all IR graphs) represents the ATR-FTIR spectra of pure chitosan powder, where typical bands can be observed, such as N–H and O–H stretching in the range of 3328–3347 cm^−1^. The evidence of left acylated groups was seen at 1657 cm^−1^ (C=O stretching of the amide I) and N-H bending, with a combination of C–N stretching vibrations at 1588 cm^−1^, while the band at 1150 can be attributed to the C–O–C bridge. The bands at wavenumbers 1066 cm^−1^ and 1026 cm^−1^ are assigned to C–O stretching [13]. The spectra in Figure 1a,b (labeled as blue) are shown for the pure thyme. At higher wavenumbers, i.e., 3450 cm^−1,^ this vibration corresponds to –OH phenolic groups in the thyme polyphenol. –CH_2_ functional groups were also observed, similar to the others. The band at 1710 cm^−1^ corresponds to the C=O bond, the band at 1588 cm^−1^ corresponds to N-H bending, at 1390 cm^−1^ C-C stretching was observed from the phenyl groups, and the band at 1260 cm^−1^ belongs to C-O stretching and, at around 1090 cm^−1,^ to the C–N stretch [28,29]. After encapsulation of the thyme into chitosan NPs and further deposition on the PE foil, the present bands from CS and THY together imply the successful functionalization onto previously activated PE foil. This can indeed be inferred from the bands that, after CSNP-THY application, appear to the PE foil (Figure 1a). Specifically, the occurrence of the N–H, O–H, –CH_2_, C–O and other functional groups, suggest the introduction of CSNP with THY onto PE foil. The same strategy was also observed after CSNP-THY deposition onto PP foil (Figure 1b). The spectrum for the pure rosemary shows typical functional groups for this kind of phenolic compounds [30]. Although the majority of the functional group overlapped with CS and PE onto PE-2%CS-CSNP ROS, the appearance of the typical band at 1460 cm^−1^ for rosemary after formulation deposition onto PE foil, the corresponding C–O stretching vibrations (amide) and C-C stretching from phenyl group [28] clearly indicate the successful application of the formed formulation with incorporated rosemary onto PE foil. As the same formulation was also applied onto PP foil, characteristic peaks originating from different species that were also present at spectra PP-2%CS-CSNP ROS, confirm the presence of the formed formulation onto PP foil (Figure 1d). The characteristic fingerprints for the cinnamon were mostly present in the range of 1800–600 cm^−1^ and are shown in Figure 1e (labeled with violet color) [30]. The peak around 1620 cm^−1^ corresponds to the stretching vibration of an aldehyde carbonyl C=O, and also the peak at 1451 cm^−1^ is characteristic for the alcohol C–OH. The cinnamon peaks at around 1110 cm^−1^ and 1070 cm^−1^ are attributed to the stretching vibrations of C–O and C–OH deformation vibration [30,31]. After application of the 2%CS-CSNP CIN onto the PE foil, typical peaks from CS and from CIN that were present after deposition indicate the presence of all the deposited compounds. Indeed, the same was also revealed for the cinnamon–chitosan-based formulation onto PP foil.

#### 2.2.2. XPS Analysis

The surface chemical composition (at %) determined by XPS of UV/O_3_ treated polymer foils and foils with the bilayer coatings based on chitosan macromolecular solution as the first layer and chitosan nanoparticles with embedded thyme, rosemary, and cinnamon, respectively, as the second layer, are shown in Table 4, Table 5, Table 6, Table 7, Table 8 and Table 9. Polymer foils with coatings of chitosan nanoparticles and each extract alone were also studied as references.

As shown from previous work [13], for the untreated (pristine) PE and PP foils mostly carbon was detected, which is in agreement with the chemical composition of the virgin foils. Small amounts of oxygen (approx. 1 at. %) were present due to the surface contamination, as often observed in XPS spectra measured on pure polymer foils. After oxygen plasma treatment, which was used there, the oxygen concentration increased to 13.7 at. % and 14.7 at. % for PE and PP, respectively [13]. Similar, and even better, could be seen here by UV/O3 activation of foils, where the oxygen concentration increased to 19. 4 at. % and 15.1 at. % for PE and PP, respectively. Higher oxygen concentration on a PP sample can, again, be explained by oxidation of the side methyl group, which is absent on PE. Increased oxygen concentration indicates clearly the introduction of oxygen—functional groups that may act as binding places for chitosan with embedded polyphenols. This is also proven by the presence of C–O, C=O and COOH groups, as seen from the high resolution XPS spectra shown in Figure 2.

When thyme was coated onto foils’ surfaces, the amount of C was obviously decreased due to covering the foils’ surfaces that consist mainly of C. The oxygen was increased whilst it was present in a big amount in the thyme structure. Pure thyme consists of carbon, hydrogen and oxygen, but because the extract does not consist of pure substances, some additional elements may be introduced through extractions, such as nitrogen, sulphur, silicium, etc. Those elements are quite often present in plant materials (plant tissue). Thus, these additional new elements of thyme extract origin were introduced onto foils’ surfaces as well (see Table 4). To clarify this, the detection of pure thyme elemental structure on a silicon wafer support was done (Table 4) to understand the contribution of thyme elements on foil. It can be seen that the presence of N, Si, Cl, S, and K originated from thyme extract, as was speculated above.

Table 5 proves the attachment of chitosan nanoparticles with embedded thyme onto foils’ surfaces due to the higher amount of detected N on the foils’ surfaces. This may also be proved by the detection of additional elements such as Na and P that belong to nanoparticles’ structure, through synthesis of which is based on crosslinking of chitosan by sodium tripolyphosphate TPP. Other present elements are of thyme origin, as discussed above.

Similar as for thyme, for foils with a rosemary coating (Table 6), the amount of C was decreased, but to a much smaller extent than for foils with thyme extract. The oxygen increased due to its presence in rosemary extract chemical structure (phenolic and acidic groups). N and Si are again present due to the extract complexity. Obviously is not possible to produce very pure extract, but there are also some other components from plants. As mentioned, the incorporation of silica within the plant cell wall has been well documented by botanists and material scientists [32], and it was already detected in rosemary extracts.

The introduction of N in high concentrations, as well as Na and P in a small amount, are evidence for chitosan–rosemary nanoparticles’ attachment (Table 7). Again, the elements of rosemary origin are present on the functionalized foils too.

From Table 8, where the surface chemical composition of the reference foils and foils treated by cinnamon coating is presented, again, decrease of C is seen, with simultaneous increase of O and the introduction of N and Si. The same happened for both the other two extracts; i.e., thyme and rosemary, and the reasons were already discussed above.

In Table 9, for the PE-2%CS-CSNP CIN and PP-2%CS-CSNP CIN, nitrogen, in similar concentration as for other bi-layered coatings systems, appeared on the foils’ surfaces. This is due to the presence of amino groups in the chitosan backbone. Na and P corroborated the nanoparticles’ structure (ionic gelation of chitosan with TPP).

XPS results for all samples show clearly that the amount of nitrogen originating from the coating was significantly increased, which means that chitosan nanoparticles adsorbed onto UV/O_3_-treated foils to a higher extent. Embedded extracts were also available on foils’ surfaces (as encapsulation efficiency pointed out), whilst typical elements of those plant extracts were detected (S, Si, Cl).

#### 2.2.3. Goniometry

Water static contact angle (SCA) results determined the hydrophilicity or hydrophobicity of PE and PP surfaces (Figure 3 and Figure 4 and Table 10). Reducing the water contact angle is of great importance for practical use, since a hydrophilic surface of the packaging foils reduces the probability of dew condensation on the surface of the foil (anti-fog efficiency) in contact with food, which worsens the packaging conditions and, thus, increases food contamination [13]. It is, thus, extremely important to avoid this process and follow the hydrophilic/hydrophobic character of foils’ surfaces. The latter is also extremely important for the active interface formation with food, whilst hydrophilicity and hydrophobicity also influence microbial inhibition. It has also been observed frequently that hydrophilic, high surface energy materials are less prone to bacterial adhesion; i.e., it is generally admitted that hydrophilic surfaces in contact with media containing organic molecules such as proteins, oppose the formation of a conditioning film harboring adhesion sites for bacteria – limiting specific adhesion/attachment of bacteria and subsequent biofilm development [33,34].

The reference PE and PP foils show contact angles of 93° and 101°, respectively, clearly declaring that these foils behave hydrophobically. After UV/O_3_ activation, the contact angles decreased by 40–50% for PP and PE, thereby rendering the polymer surfaces hydrophilic. With the functionalization of the foils using chitosan-extract nanoparticles, the SCA increased, on average, in comparison to the activated polymer foils PE-UV/O_3_ and PP-UV/O_3,_ but remained below 90°, which is, thus, still considered hydrophilic. The decrease of contact angles occurred in the following order for functionalized foils: a) PE: PE-2%CS-CSNP CIN > PE-2%CS-CSNP ROS > PE-2%CS-CSNP THY and for b) PP: PP-2%CS-CSNP ROS > PP-2%CS-CSNP CIN > PP-2%CS-CSNP THY. In both cases; among functionalized foils, these foils coated by 2%CS-CSNP THY showed the lowest SCA, which is in accordance with the measured surface tension of dispersions that show the lowest value for the CSNP THY. This value accounts for this system of 39.52 mN/m, whilst for CSNP-CIN and CSNP-ROS they are 46.27 mN/and 43.58 mN/m, respectively.

#### 2.2.4. Surface Zeta Potential

Surface charge is a fundamental parameter for the enhancement or suppression of the interaction between dissolved compounds in an aqueous solution and solid material surfaces. The zeta potential (ζ), used primarily as the indicator for solid surface charge, is an appreciated parameter for the comparison of material surfaces before and after surface treatment, as well as their charging behavior in an aqueous solution [35]. Figure 5 shows the pH dependence of the zeta potential for pristine and UV/O_3_-activated PE foils, and for PE functionalized with chitosan and chitosan-extract nanoparticles.

As was previously reported, the surface of the untreated PE behaves inertly in the presence of an aqueous solution of a 1:1 electrolyte such as KCl [36]. However, the pH dependence of its zeta potential differed significantly from 0 mV. Above the isoelectric point (IEP i.e., pH of the aqueous solution where ZP = 0 mV) at approx. pH 4, the PE-water interface was negatively charged, whereas at low pH, the sign of this interfacial charge reversed to positive. This behavior is typical for a polymer or any other hydrophobic material surface without any surface functional groups [37,38]. The non-zero zeta potential and the IEP at pH 4 were determined by the adsorption of water ions at hydrophobic surfaces. Above pH 4, the adsorption of hydroxide ions dominated and generated a negative interfacial charge, which was detected by the zeta potential [38]. At pH 4, the concentration of adsorbed OH^−^ ions equaled that of H_3_O^+^ ions. Below pH 4, H_3_O^+^ ion adsorption dominated, thereby rendering the interfacial charge positive.

Surface activation of the PE films by UV/O_3_ treatment had an effect on the zeta potential. The magnitude of the negative zeta potential increased well, and the IEP shifted to a more acidic pH. This suggests that UV/O_3_ treatment introduced polar groups, which confirms the increase of the relative atomic oxygen percentage observed by XPS. Moreover, a cleaning effect from organic non-polar contaminants and changes in the foil morphology were expected, which contributed to the higher negative zeta potential and the shift in the IEP. A similar effect on the surface zeta potential was observed for a poly (ethylene terephthalate) PE foil activated by O_2_- and CO_2_- plasma treatment, respectively [38]. Foils which were coated by the 2% macromolecular chitosan solution (as the first coating layer), PE-2%CS, show clearly the successful attachment of chitosan onto the PE foil. The IEP at pH 8.3 indicates the full coverage of PE by the chitosan macromolecules. An IEP at pH 8.2 was reported for a chitosan nanofiber nonwoven [39]. Behary et al. also reported a similar result, but for a polyester nonwoven coated by chitosan [40]. Compared to pristine and UV/O_3_-activated PE, a much higher positive plateau at ζ = +20 mV was monitored for this sample. A similar behavior was observed for the PE foil coated by a 2% chitosan macromolecular solution, and, furthermore by pure chitosan nanoparticles (PE-2%CS-CSNP). The aim was to see any additive effect of this differently structured chitosan on a foil cationic behavior. The IEP was slightly shifted to the more alkaline region, and a minor increase was observed in the positive plateau. Obviously, nanoparticles provide higher specific surface area and, thus, a higher content of available protonated amino groups, resulting in the shift of the IEP, as well as in the increase of the positive ZP.

The negative ZP values at a pH > IEP for all chitosan-treated foils can be explained by the presence of the remaining acetyl groups in chitosan (the degree of acetylation was ≈90%). For all samples coated with a bilayer of chitosan and chitosan nanoparticles with embedded extracts, the shift of the isoelectric point to lower pH occurred due to the introduction of the acidic character of the extracts and the partly/blocking of amino groups by the chemical reaction of extracts (polyphenols) with chitosan. It was shown in our previous work that the attachment of antioxidants onto chitosan fibres blocked accessible chitosan amino groups, due to a pronounced interaction between these amino groups with the OH groups of flavonoids [27]. Similar interactions between amino groups of chitosan and phenolic groups of extracts may have occurred here. If extracts were not only embedded inside, but also attached to the outer part of chitosan nanoparticles, the acidic character of their phenolic groups would still be present at the surface of functionalized foils. Moreover, with the release of extracts from chitosan nanoparticles, which was enforced when lowering pH, the presence of these extracts was monitored through a shift in the IEP to the more acidic region. This may be expected from a comparison with the results of the total phenolic group content (Table 3) represented by the gallic acid (GA) equivalent for phenolic acids in cinnamon, rosemary, and thyme extract solutions. The assumption of the accessibility of acidic extracts is also supported by the increase in the negative zeta potential. Assuming that the foils are fully covered by the first layer of chitosan macromolecules, the presence of a higher negative plateau in the zeta potential after coating with chitosan nanoparticles with embedded extracts is explained by the introduction of extracts to the foil surface. However, the shifts of the isoelectric points for samples PE-2%CS-CSNP ROS, PE-2%CS-CSNP THY, and PP-2%CS-CSNP CIN are not in accordance with the content of gallic acid (amount of phenolics) for each extract. This suggests a non-uniform and different distribution of extracts in chitosan nanoparticles. Such a surface heterogeneity affects the availability of extracts at the surface and, consequently, the presence of the polyphenol quantity.

Figure 6 shows the pH-dependence of the four pristine and UV/O_3_-activated PP foils. The pristine PP foil shows a typical polyolefin character with the isoelectric points at pH 3.8 and an almost linear dependence of the zeta potential on pH. The pronounced linear dependence of the zeta potential on pH, and the more negative zeta potential at pH > 6.5, reflects a higher hydrophobicity for the pristine PP foil as compared with the pristine PE foil in accordance with the water contact angles (Table 10). The effect of UV/O_3_ activation on the zeta potential was similar for PP and PE. The isoelectric point of PP shifted from pH 3.8 to pH 3.5 because of the increase in oxygen-containing groups, like hydroxyl, aldehyde, ketone, and carboxylic groups, on the foil surface (as shown with XPS-high resolution spectra in Figure 2). The increase in the negative zeta potential was also in accordance with the introduction of such groups, as well as with the cleaning effect of the PP foil surface. Despite the higher hydrophobicity of PP, which suggests an enhanced inert behavior, the activation by UV/O_3_ treatment introduced an even stronger effect for PP than for PE (e.g., at pH 5, ζ = −35 mV for UV/O_3_-treated PP but ζ = −25 mV for activated PE). Different to the pristine foils, the zeta potential contradicted the corresponding water contact angles. We explain this discrepancy by a shorter durability of the surface activation for PP. Proof of this conclusion requires further analyses of the time-dependence of surface activation of these polyolefins.

As for PE, a significant shift of the IEP towards the alkaline range was observed after the adsorption of chitosan macromolecules (PP-2%CS) and the subsequent coating by chitosan nanoparticles (PP-2%CS-CSNP). Between both samples, PP-2%CS and PP-2%CS-CSNP, there was a small difference in the IEPs in the range of pH 7.5-7.8, but almost the same positive plateau ZP appears. The zeta potential again proves the introduction of functional amino groups of chitosan on the PP foil surface. However, the lower IEP compared to chitosan-functionalized PE suggests an incomplete surface coverage. The conclusion on a lower surface coverage may be supported by the more negative ZP plateau of UV/O_3_-activated PP in comparison to UV/O_3_-treated PE, and by the higher water contact angle. However, the zeta potential of PP-2%CS, especially at pH > IEP, indicates that the surface and interfacial charge was dominated by chitosan. Only a small shift in the IEP to lower pH was observed after adsorption of chitosan nanoparticles with embedded extracts on PP-2%CS-coated PP foils. In addition, at pH > IEP, the zeta potential shifted to more negative values, and at pH < IEP, to more positive values as compared to PP-2%CS. The higher negative zeta potential at high pH indicates the presence of acidic polyphenol groups of the extracts. If these groups were distributed on the PP surface, they were available on the external part of the nanoparticles, or got released from the interior of the CSNPs. Almost the same IEP as for the PP-2%CS and the same negative zeta potential at pH > IEP was observed for PP-2%CS-CSNP CIN. This means that CIN did not affect the presence of chitosan, and was obviously embedded in the inner part of the nanoparticles. For PP-2%CS-CSNP ROS and PP-2%CS-CSNP THY, the IEP shifted to pH 6 and pH 6.3, respectively. Their negative ZP values were almost the same and lower than the ZP for the foils coated by chitosan macromolecules only. When comparing the CS-extract-coated foils of PP and PE, one can conclude that less content of polyphenols is available on the PP surface. The difference in the zeta potential is also explained by a lower desorption of the extract coating during the measurement, suggesting a stronger adhesion of CS-extract NPs to the UV/O_3_-activated PP surface.

To summarize, the zeta potential results for both PP and PE foils proved a successful functionalization with CS and CS NP with embedded extracts. The zeta potential of functionalized foils further confirmed the presence of bioactive, amino and polyphenol groups on the surface, which, in packaging applications, are prone to contact with the food, and are the driving force for the bioactivity of functionalized foils, thus prolonging the shelf life time.

#### 2.2.5. SEM

Figure 7 presents SEM images of the surface morphology for uncoated (pristine and UV/O_3_-activated) PE and PP foils. Further images are also shown of UV/O_3_-treated foils coated by chitosan in a first layer and CSNP–extract formulations in a second (upper) layer. For non-treated PE and PP foils a rough surface with some defects or impurities on the surface were seen, that may have originated either from contaminated foils or impurity that may have been introduced during the foil preparation for the analysis. When foils were treated with UV/O_3_ activation, quite smooth and clean surfaces were present on the images, especially in the case of PP-activated foil. It is known that UV/O_3_ clean the surface, which is reflected in foils’ surface morphology in a smoother manner [41]. For both PP and PE foils coated by 2%CS-CSNP THY, a rough coating appeared with visible particles on the surface which formed cross-linked or branched agglomerates and covered the foil surface uniformly. The latter was revealed at the higher magnification. However, in the case of the PP-2%CS-CSNP THY, the presence of NPs with diameter less than 1 μm could be seen clearly, whereas, in the case of the PE-2%CS-CSNP THY, they could be partly embedded into the previously applied layer of chitosan. On the contrary, very different morphology could be obtained from the foils deposited with the CSNP having embedded ROS. While lower magnification revealed the difference in contrast for the covered and uncovered foil places with applied formulations, the higher magnification in both cases revealed that brighter areas were composed of the individually coated particles, ranging from 100 nm to less than 1 μm. In the case of the PP-functionalized foil, the adhesion of the particles was more improved than in the case of the PE-functionalized foil. In fact, the same manner of the larger density of particles can be seen in all cases of the PP-functionalized foil. Moreover, in samples PE-2%CS-CSNP ROS and PP-2%CS-CSNP ROS, a smoother surface coating was found, with some small amount of particles present with a dimension of 0.2 μm. When taking a closer look into the morphology of the foils modified with the 2%CS-CSNP CIN, the effect of the different foil could be seen. The PE-functionalized foil consisted of smaller particles well below 1 μm, that seemed to be partly embedded into the first applied layer of chitosan film. Very different sizes of the CSNP CIN could be seen on the PP-functionalized foil, where some particles exceeded 1 μm in size and showed a larger degree of the heterogeneity. From all of these comparisons, one can conclude on partly covered surfaces (these can be seen mostly through the difference in contrast, where brighter areas correspond to deposited particles), by PE-2%CS-CSNP CIN the coatings distributed on foils’ surfaces were the most homogenous and smoother, uniformly covering the foil with less observed agglomerates. This is somehow surprising, whilst 2%CS-CSNP CIN dispersion showed a PDI index of 1.000, and due to the size heterogeneity of particles less homogeny coating of films was expected, with the potential appearance of wide size distribution particles on the surface. However, this was the case for sample PP-2%CS-CSNP CIN. From the SEM images successful application of different formulations onto PP/PE foils in the form of the particles can be concluded that, in some cases, showed spherical morphology with the particle’s size around 1 μm.

It seems that quite a high amount of coatings and good covering of foils was obtained in most of the cases by foils’ functionalization and, thus, good bioactive properties in interface with food could be expected. The obtained results will, in a subsequent paper, be related to the antimicrobial, antioxidant activities and barrier properties of foils, determined by the Standard ASTM E2149 method.

## 3. Materials and Methods

### 3.1. Materials

The following chemicals were used during this research: low-molecular weight (LMW) chitosan (50 to 190 kDa), deacetylated chitin, Poly (D-glucosamine) from Sigma-Aldrich; sodium tripolyphosphate—TPP (MW = 367.85 g/mol) from Acros Organics, Belgium; hydro-alcoholic natural tincture of thyme (*Thymus vulgaris* L.), 10% *T. vulgaris* L., 90% ethanol (15%) from Soria Natural, Spain; acetic acid (MW = 60.05 g/mol), ≥99.8% from Sigma-Aldrich; ethanol (MW = 46.07 g/mol), 99.8% (GC) from Honeywell Sigma-Aldrich; MilliQ water—Milli-Q Direct system—Millipore, 0.2 μm PES High Flux Capsule Filter; polyethylene—PE normal quality, transparent, GSM = 46.28 g/m^2^ (thickness 50 µm, slippery 0.207) from Makoter d.o.o.; polypropylene—PP normal transparent oriented, GSM = 22.93 g/m^2^ (thickness 27 µm, slippery 0.278) from Manucor S.p.A., Italy.

#### Extraction of Plant Extracts

Soxhlet extraction: approximately 20 g of ground material (thyme—leaves, rosemary—leaves, cinnamon—bark,) was poured into a flask. Before grinding, materials were dried to a constant mass. 150 mL of solvent (EtOH) was added to each ground material. The flask was then placed in an ultrasonic bath, where it was left for about an hour. After extraction was complete, the obtained extract solution was filtered, and collected in a pre-weighed round bottom flask. Finally, the solvent was evaporated under vacuum at 40 °C to collect the extract.

Cold solvent extraction: the powdered above prepared materials (20 g) were extracted by stirring, using a magnetic stirrer with 150 mL of EtOH at 25 °C for 4 h. The extract was filtered for removal of solid particles. The extracts were cooled to room temperature and concentrated under vacuum at 40 °C.

### 3.2. Preparation of Solutions/Dispersions

Aqueous chitosan solutions with 1% and 2% (*w*/*v*) were prepared by dissolving LMW chitosan powder in MilliQ water. Glacial acetic acid was added drop-wise to the chitosan solution during constant magnetic stirring to enable chitosan dissolution. Solutions were left stirring for 24 h until a homogeneous dispersion was obtained. The pH was adjusted to 4.0.

Extract solution: all extracts, prepared as pointed out in the description of the extraction procedure, were dissolved in absolute ethanol. The concentration of the extract was determined according to a determined minimal inhibitory concentration (MIC), therefore 20 mg/mL cinnamon and rosemary solutions were prepared, and the concentration of thyme was 10% (*v*/*v*).

TPP solution: as a crosslinking agent, sodium tripolyphosphate (TPP) was used. TPP powder was dissolved in MilliQ water, the concentration of the solution was 0.2% (*w*/*v*) in order to obtain 5:1 chitosan to TPP ratio (the optimal ratio according to a previously published paper [42]).

Preparation of chitosan nanoparticles (CSNP) with incorporated thyme/rosemary/cinnamon extract: CSNP embedded with thyme/rosemary/cinnamon extract were synthesized by the ionic gelation method. 20 mL of TPP solution and 10 mL of extract solution were added drop-wise to 20.0 mL of 1% CS solution during stirring. The solution was submitted to continuous stirring for 1h at room temperature, and the particles were formed spontaneously. The pH of the solution was adjusted to 4.0 using acetic acid. The final concentration of the extract in the solution was 4 x its MIC in order to obtain good antimicrobial activity.

### 3.3. Functionalized Foils

To achieve better adhesion of chitosan solutions onto the foils, PE and PP were activated by Novascan UV/OZONE cleaner. The foils were first cleaned and dried, and then exposed to UV/O_3_ surface treatment, PE for 15 min and PP for 20 min at 25 °C.

Coating was applied in two layers. The first layer was a chitosan macromolecular solution (2%,) and the second layer was a dispersion of CSNPs with incorporated thyme, rosemary or cinnamon extract. The method for coating application was roll-to-roll printing, using a Johannes Zimmer machine, Austria. After each layer, the foils were dried at room temperature. Chitosan macromolecular solution was applied as the first layer, due to its high antibacterial efficacy and for better adhesion of extracts’ solutions, which exhibit high antioxidant and antifungal activity. A sample description is given in Table 11.

### 3.4. Methods

#### 3.4.1. Particle Size, Polydispersity Index PDI and Zeta Potential Determination

The hydrodynamic diameter, the polydispersity index (PDI), and the zeta potential of nanoparticle dispersions were determined by dynamic and electrophoretic light scattering, using a Zetasizer Nano ZS (Malvern Instruments Ltd., UK). Before being analyzed, the dispersion was stirred for 15 min and adjusted to pH 4 with acetic acid. For carrying out the DLS measurements, the sample was filled up to 1 cm in a disposable cuvette. A disposable folded capillary cell with electrodes was used for the zeta potential determination in aqueous dispersions. Data were collected using the Zetasizer software.

#### 3.4.2. Encapsulation Efficiency

The encapsulation efficiency was determined for chitosan nanoparticles containing thyme extract, rosemary extract, and cinnamon extract, respectively. The encapsulation efficiency, or entrapment efficiency, was determined as the ratio between the concentration of the extract incorporated in the nanoparticles and its initial concentration. The concentration of the incorporated extracts was determined indirectly by spectrophotometric concentration monitoring of the non-incorporated substance in the supernatant after the centrifugation of the nanoparticle dispersion. The monitoring was performed by the maximum wavelength (285 nm, 285 nm, and 282 nm) of each of the incorporated substances (i.e., thyme extract, rosemary extract and cinnamon extract). The encapsulation efficiency EE was evaluated using Equation (1):(1)$$EE=\frac{C_{\mathrm{NP}}}{C}=\frac{C-C_{\mathrm{SUP}}}{C}\times 100\%$$
where *C* is the initial concentration of extracts used for the preparation of the nanoparticle dispersion (g/L), *C*_NP_ is the concentration of incorporated extracts in nanoparticles (g/L), and *C*_SUP_ is the concentration of substances in the supernatant (g/L).

#### 3.4.3. Total Phenols’ Content

The total phenolic content of the extracts was determined using a Folin–Ciocalteu reagent, as described in the literature [43]. Briefly, the Folin–Ciocalteu reagent solution was prepared by diluting the stock solution with distilled water in a ratio of 1:10. 50 mg of the extract was weighed in a 10 mL volumetric flask, and diluted with MeOH. 2.5 mL of the Folin–Ciocalteu reagent solution, and 2 mL of 0.075 g/mL Na_2_CO_3_ were added to 5 mL of the prepared extract solution. The mixture was left for 30 min at room temperature (25 ± 2 °C), then the absorbance of the solution was measured at 765 nm using a UV-visible spectrophotometer. The total phenolic contents were determined in triplicate for each sample. The calibration curve of gallic acid (GA) was used for quantification of the total phenolic compounds, and the amount of phenolic compounds in the samples was expressed as gallic acid equivalents, in mg of GA/g of material.

#### 3.4.4. Surface Tension Measurements

The surface tension of all dispersions was determined employing a Krüss (Hamburg, Germany) K-12 tensiometer and the Wilhelmy plate method, which was cleaned with distilled water and acetone and flame-dried before each measurement. Overall, each surface tension value reported was the average of 10 measurements. Before measuring the surface tension, the samples were stirred in a thermostated vessel that was closed to prevent evaporation. The measurement vessel was connected to a controlled Julabo thermostat-cryostat bath.

#### 3.4.5. ATR-FTIR Spectroscopy

The ATR-FTIR spectra were recorded on a Perkin Elmer Spectrum GX NIR FT-Raman spectrometer. The ATR accessory contained a diamond crystal. All spectra (16 scans at 4 cm^−1^ resolution, background, and the sample spectra, were obtained in the 400–4000 cm^−1^ wavenumber range) were recorded at room temperature. Spectra of samples were deconvoluted with a smoothing filter and baseline corrected (automatic).

#### 3.4.6. X-ray Photoelectron Spectroscopy (XPS) Analysis

XPS spectra were recorded using a PHI TFA XPS from Physical Electronics, USA, in order to assess the surface of the samples. The base pressure in the XPS analysis chamber was approximately 6 × 10^−8^ Pa. The samples were excited with X-rays over a 400-µm spot area with monochromatic Al Kα_1,2_ radiation (1486.6 eV) operating at 200 W. Photoelectrons were detected with a hemispherical analyser, positioned at an angle of 45° with respect to the sample surface. The energy resolution was about 0.6 eV. Spectra were recorded from at least three locations on each sample, using an analysis area of 400 μm. Surface elemental concentrations were calculated from the survey-scan spectra using the Multipak software.

#### 3.4.7. Goniometry

Water contact angles were measured using a goniometer (DataPhysics, Germany) to estimate the surface wettability of coated foils. The static contact angle of Milli-Q water was determined in triplicate using a drop volume of 3 µl for each measurement.

#### 3.4.8. Surface Zeta Potential

Zeta potential analyses of pristine and CSNP-coated polymer foils were carried out using an Anton Paar SurPASS 3 instrument, equipped with an adjustable gap cell. A pair of each foil sample was mounted on sample holders with a cross-section of 20 mm x 10 mm using double-sided adhesive tape. The sample holders were then inserted in the measuring cell, and the distance between foil surfaces was adjusted to 110 ± 10 µm. A 0.002 M KCl solution was used as the electrolyte, the initial pH was adjusted to pH 10 with 0.05 mol/l KOH, while, during automatic titrations, changes in pH (from about pH 10 to pH 2) were achieved by the addition of 0.05 M HCl. The zeta potential was calculated from streaming potential measurements using the equation by Helmholtz and Smoluchowski [44].

#### 3.4.9. Scanning Electron Microscopy, SEM Analysis

For scanning electron microscopy (SEM) imaging of the surface morphology and for the conformation of a successful particle deposition, samples of pristine and functionalized PE and PP foils were cut into small pieces (approx. 0.5 × 0.5 cm^2^). Afterwards, these pieces were attached to the aluminum sample holders with an adhesive carbon tape in order to ensure conductivity. A Carl Zeiss Supra 35VP scanning electron microscope was employed, with an accelerating voltage of 1 kV and a variable working distance using a 30–20 µm-sized aperture. All images were taken at same magnifications.

## 4. Conclusions

XPS and ATR-FTIR methods proved the successful binding of bilayer formulations of a 2% chitosan macromolecular solution and further chitosan nanoparticles with embedded plant extracts onto PP and PE foils. IR graphs confirmed the bands typical for specific coatings. The XPS results confirm the availability of N in the surface layer, which will create an active interface in contact with food in the packaging application. The elemental composition suggests that the amino groups of chitosan origin, primarily responsible for antimicrobial efficiency are available in the thin-film surface coating, and will be reactive at the packaging-food interface. In addition, some extracts are accessible at the external foil surface, which may contribute significantly to the antioxidant activity at the interface with food. The successful coating was also supported by the zeta potential measurements, which showed the availability of protonated amino groups onto the functionalized surfaces of foils and the accessibility of extracts. The presence of both chitosan and polyphenols (of the extracts) is extremely important to achieve foils active concept. For the functionalized foils, the initial hydrophobic character was converted to hydrophilic, confirmed by a decrease in the water contact angles. The latter is also important for anti-fog foil properties, which reduces food perishability.

From the SEM analysis, it may be concluded that a high amount of coatings with a good and quite homogeneous coverage of the foil surface was obtained by the functionalization.

The study was of a great importance, whilst when developing concepts for active packaging based on coatings, it is extremely important to get insight into the surface chemistry, since it is responsible for the response of surface activators at the interface with food. Due to foils characteristics good bioactive properties at the interface with food could be expected, which will be discussed in a subsequent paper.

## Figures and Tables

**Figure 1 ijms-21-00495-f001:**
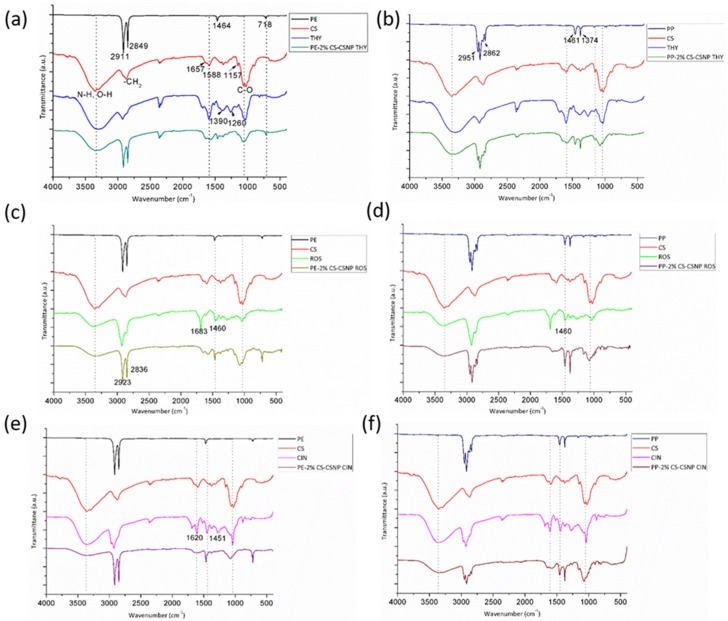
ATR-FTIR spectra of the pure polyethylene (PE), polypropylene (PP), chitosan (CS), and individual antioxidants, together with the applied formulation of thyme onto PE (**a**) and PP foil (**b**), rosemary onto PE (**c**) and PP foil (**d**); and cinnamon onto PE (**e**) and PE (**f**) foil.

**Figure 2 ijms-21-00495-f002:**
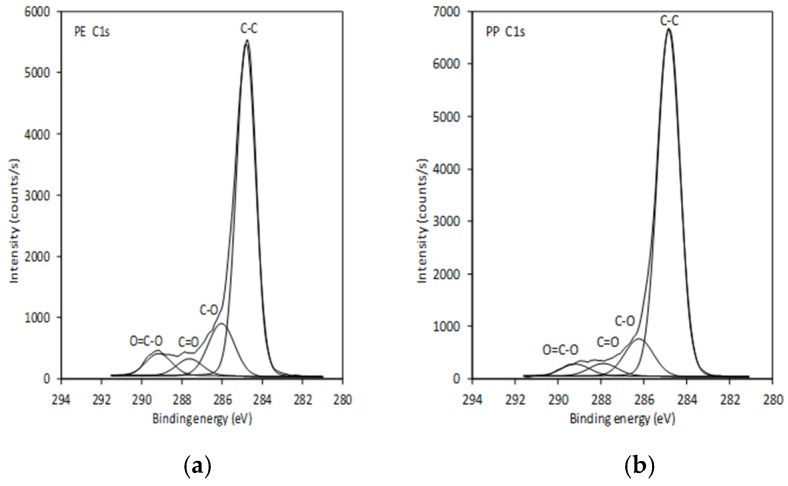
High resolution XPS spectra for: (**a**) PE-UV/O_3_; (**b**) PP-UV/O_3_ pretreated samples.

**Figure 3 ijms-21-00495-f003:**
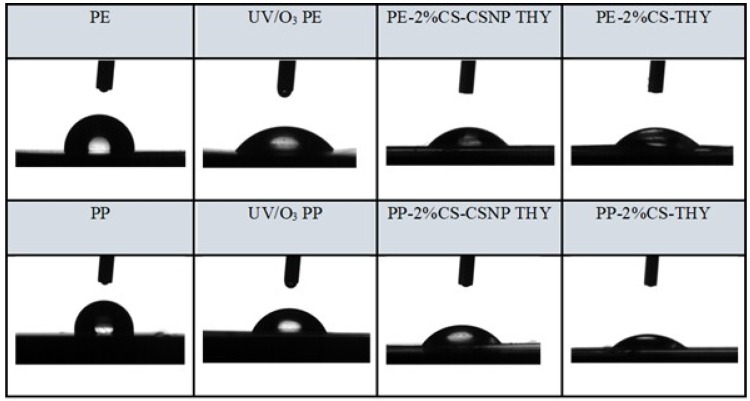
Contact angle (SCA) of foils functionalized by thyme extract (THY) in combination with chitosan.

**Figure 4 ijms-21-00495-f004:**
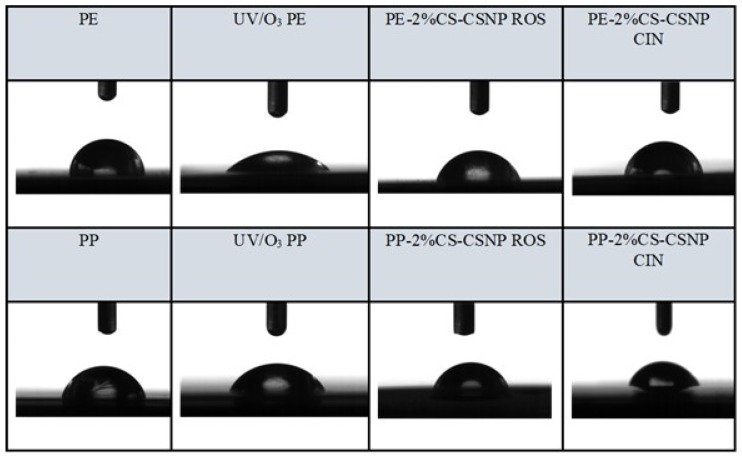
Static contact angle (SCA) of foils functionalized by cinnamon extract (CIN) and rosemary extract (ROS) in combination with chitosan.

**Figure 5 ijms-21-00495-f005:**
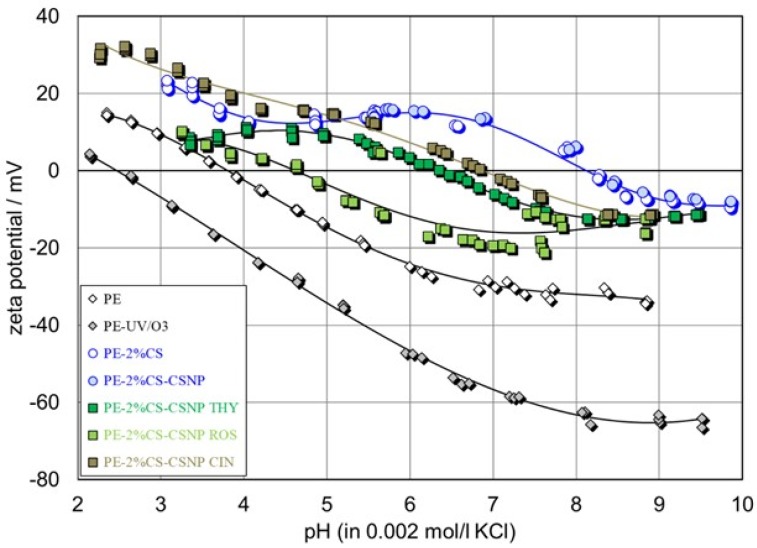
Zeta potential as a function of pH of PE and PE functionalized foils.

**Figure 6 ijms-21-00495-f006:**
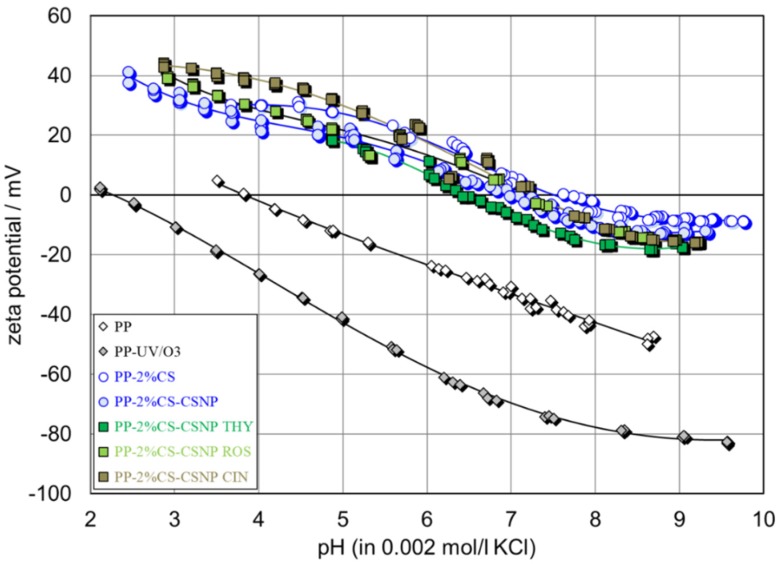
Zeta potential as a function of pH of PP and PP functionalized foils.

**Figure 7 ijms-21-00495-f007:**
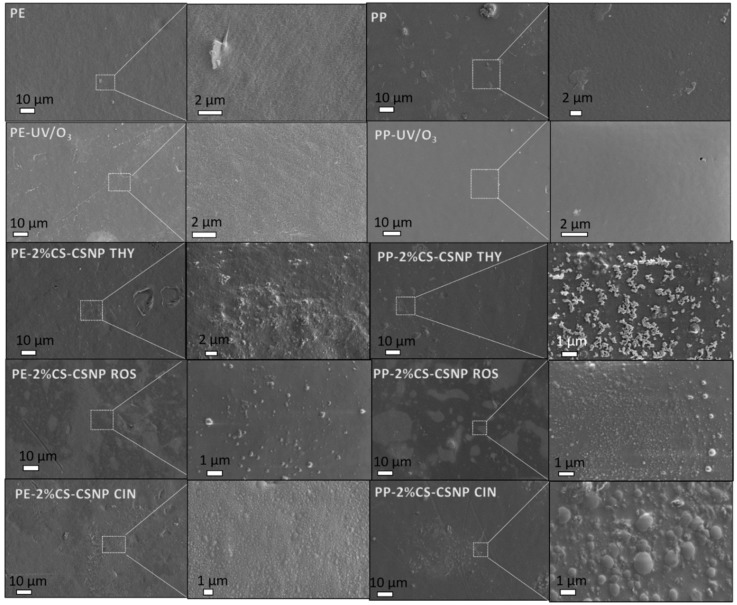
SEM images of functionalized foils.

**Table 1 ijms-21-00495-t001:** Size (hydrodynamic diameter), polydispersity index (PDI) and zeta potential (ZP) of particle dispersions at pH 4.

Dispersion	Size (nm)	PDI	ZP (mV)
CSNP	293 ± 80	0.871	37 ± 2
CSNP THY	1138 ± 93	0.917	39 ± 3
CSNP ROS	6627 ± 353	0.642	39 ± 5
CSNP CIN	2993 ± 435	1.000	40 ± 4

**Table 2 ijms-21-00495-t002:** The encapsulation efficiency of thyme extract, rosemary extract and cinnamon extract in chitosan nanoparticles.

Dispersion	Degree of Encapsulation (%)
CSNP THY	77.93 ± 2.85
CSNP ROS	83.51 ± 1.76
CSNP CIN	86.03 ± 4.25

**Table 3 ijms-21-00495-t003:** Total phenolic content of extract solutions and chitosan nanoparticles with encapsulated extracts.

Sample	Concentration of GA in Extract Solution, W_GA_ (mg/g)
THY	64.00
ROS	154.00
CIN	214.00
CSNP THY	4.50
CSNP ROS	4.31
CSNP CIN	5.38

**Table 4 ijms-21-00495-t004:** Surface chemical composition of pristine and UV/ozone (UV/O_3_)-activated polymer foils and of foils coated with thyme (at. %).

Sample	C	N	O	Si	S	Cl	K
PE-UV/O_3_	80.3	/	19.4				
PP-UV/O_3_	84.9	/	15.1				
PE-THY	67.0	1.4	29.4	0.8	0.1	0.4	1.0
PP-THY	63.7	1.1	32.1	1.7	0.1	0.3	1.0
Thyme coated on silicon wafer	69.9	0.8	27.6	0.1	0.1	0.6	1.1
Difference (%) * PE	−13.3	1.4	10	0.8	0.1	0.4	1.0
PP	−21.2	1.1	17	1.7	0.1	0.3	1.0

* The difference is calculated between functionalized and reference PE/PP-UV/O_3_ treated sample.

**Table 5 ijms-21-00495-t005:** Surface chemical composition (at. %) of the reference foils and foils treated by the two-layer coating: 2%CS-CSNP THY (at. %).

Sample	C	N	O	Na	Si	P	S	Cl
PE-UV/O_3_	80.3	/	19.4					
PP-UV/O_3_	84.9	/	15.1					
PE-2%CS-CSNP THY	62.2	4.1	31.5	0.3	1.2	0.6	0.1	0.1
PP-2%CS-CSNP THY	59.3	4.8	33.4	0.1	1.0	1.1	0.2	0.1
Difference (%) PE	−18.1	4.1	12.1	0.3	1.2	0.6	0.1	0.1
PP	−25.6	4.8	18.3	0.1	1.0	1.1	0.2	0.1

**Table 6 ijms-21-00495-t006:** Surface chemical composition of the reference foils and foils treated by rosemary coating (at. %).

Sample	C	N	O	Si
PE-UV/O_3_	80.3	/	19.4	
PP-UV/O_3_	84.9	/	15.1	
PE-ROS	81.1	0.6	17.9	0.4
PP-ROS	75.4	0.3	22.7	1.6
Difference (%) PE	0.8	0.6	−1.5	0.4
PP	−9.5	0.3	7.6	1.6

**Table 7 ijms-21-00495-t007:** Surface chemical composition (at. %) of the reference foils and foils treated by the two-layer coating: PE-2%CS-CSNP ROS.

Sample	C	N	O	Na	Si	P
PE-UV/O_3_	80.3	/	19.4			
PP-UV/O_3_	84.9	/	15.1			
PE-2%CS-CSNP ROS	66.7	4.0	26.8	0.4	1.9	0.2
PP-2%CS-CSNP ROS	62.0	3.8	27.8	0.3	5.9	0.3
Difference (%) PE	−13.6	4.0	7.4	0.4	1.9	0.2
PP	−22.9	3.8	12.7	0.3	5.9	0.3

**Table 8 ijms-21-00495-t008:** Surface chemical composition (at. %) of the reference foils and foils treated by cinnamon coating.

Sample	C	N	O	Si
PE-UV/O_3_	80.3	/	19.4	
PP-UV/O_3_	84.9	/	15.1	
PE-CIN	77.1	0.5	21.1	1.3
PP-CIN	75.3	/	23.6	1.1
Difference (%) PE	−3.2	0.5	1.7	1.3
PP	−9.6	0	8.5	1.1

**Table 9 ijms-21-00495-t009:** Surface chemical composition of the reference foils and foils treated by the two-layer coating: PE-2%CS-CSNP CIN in at. %.

Sample	C	N	O	Na	Si	P	Cl
PE-UV/O_3_	80.3	/	19.4				
PP-UV/O_3_	84.9	/	15.1				
PE-2%CS-CSNP CIN	63.3	4.0	27.4	0.7	4.3	0.3	/
PP-2%CS-CSNP CIN	62.2	3.2	28.4	0.2	5.7	0.1	0.2
Difference (%) PE	−17.0	4.0	8.0	0.7	4.3	0.3	/
PP	−22.7	3.2	13.3	0.2	5.7	0.1	0.2

**Table 10 ijms-21-00495-t010:** Values of measurements for reference and functionalized foils.

Sample	Average Angle (α/°)	Difference (%)
PE	92.54°	/
PP	100.60°	/
PE-UV/O_3_	48.67°	47.4
PP-UV/O_3_	65.89°	34.5
PE-2%CS-CSNP THY	51.00°	44.9
PP-2%CS-CSNP THY	56.30°	44.0
PE-2%CS-CSNP ROS	74.85°	17.7
PP-2%CS-CSNP ROS	73.45°	27.2
PE-2%CS-CSNP CIN	82.30°	10.2
PP-2%CS-CSNP CIN	65.40°	35.2

**Table 11 ijms-21-00495-t011:** Sample description.

Sample Notation	Description of Sample
PE	polyethylene foil
PP	polypropylene foil
PE-UV/O_3_	PE foil treated with a UV/OZONE system
PP-UV/O_3_	PP foil treated with a UV/OZONE system
CS	chitosan powder
THY	thyme extract- *Thymus vulgaris* L.
ROS	rosemary extract
CIN	cinnamon extract
CSNP	chitosan nanoparticles’ dispersion
CSNP THY	chitosan nanoparticles with encapsulated thyme extract dispersion
CSNP ROS	chitosan nanoparticles with encapsulated rosemary extract dispersion
CSNP CIN	chitosan nanoparticles with encapsulated cinnamon extract dispersion
PE-2%CS-CSNP THY	UV/O_3_ treated PE foil, coated with 2% CS and CSNP THY
PP-2%CS-CSNP THY	UV/O_3_ treated PP foil, coated with 2% CS and CSNP THY
PE-2%CS-THY	UV/O_3_ treated PE foil, coated with 2% CS and thyme extract solution
PP-2%CS-THY	UV/ O_3_ treated PP foil, coated with 2% CS and thyme extract solution
PE-THY	UV/O_3_ treated PE foil, coated with thyme extract solution
PP-THY	UV/O_3_ treated PP foil, coated with thyme extract solution
PE-2%CS-CSNP ROS	UV/O_3_ treated PE foil, coated with 2% CS and CSNP ROS
PP-2%CS-CSNP ROS	UV/O_3_ treated PP foil, coated with 2% CS and CSNP ROS
PE-ROS	UV/O_3_ treated PE foil, coated with rosemary extract solution
PP-ROS	UV/O_3_ treated PP foil, coated with rosemary extract solution
PE-2%CS-CSNP CIN	UV/O_3_ treated PE foil, coated with 2% CS and CSNP CIN
PP-2%CS-CSNP CIN	UV/O_3_ treated PP foil, coated with 2% CS and CSNP CIN
PE-CIN	UV/O_3_ treated PE foil, coated with cinnamon extract solution
PP-CIN	UV/O_3_ treated PP foil, coated with cinnamon extract solution
PE-2%CS	PE foil treated with UV/O_3_, applicated with 2% CS solution
PP-2%CS	PP foil treated with UV/O_3_, applicated with 2% CS solution
PE-2%CS-CSNP	PE foil treated with UV/O_3_, applicated with 2% CS solution and CSNP
PP-2%CS-CSNP	PE foil treated with UV/O_3_, applicated with 2% CS solution and CSNP

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
