# Peer review of "Physicochemical Characterization of Packaging Foils Coated by Chitosan and Polyphenols Colloidal Formulations"

_ijms, 2020, doi:10.3390/ijms21020495_

Round 1
Reviewer 1 Report
In my opinion this paper can be published in present form
Author Response
Thank you very much for the time and effort to read our article.
With best regards
Lidija Zemljič
Reviewer 2 Report
The title of the manuscript is too long, authors may short the title. The abstract is not clear, authors need to put more effort to highlight their work more concisely. In abstract “Two representative, commercially used foils, polypropylene (PP) and polyethylene (PE), previously activated by UV/ozone, were functionalized using chitosan-extract nanoparticle dispersions”, as the functionalized particles are not in nano size, why author have used nanoparticles dispersions. Authors should mention the reasons why they have used UV/ozone? And explain the mechanism how it affects the surface of Polypropylene and Polyethylene. The size measurement performed with DLS, I am curious to know whether the size of CSNP THY 1138±93 , CSNP ROS 6627±353 and CSNP CIN 2993±435 DLS is good for below 1000nm, author have measure size 6627 with error 353nm and PDI is very abrupt, please recheck the use of DLS instrument. I am confused with sentence, “The positive zeta potential further confirms the accessibility of amino groups of chitosan, which suggests the encapsulation of the polyphenols into the chitosan nanoparticles” . As polyphenols are negative charge after coating with chitosan nanoparticles, it should reduce the charge of the particles. The overall results and discussion needs to be improved. Authors can improve conclusion which can highlight major finding and their prospective. Authors have not used any International Journal of Molecular Sciences references, As you are interested for this journal, if possible insert some related recent references from International Journal of Molecular Sciences.
Manuscript ID: ijms-677446
Type of manuscript: Article
Title: Packaging foils functionalized by colloidal synergistic formulation of polysaccharide chitosan and different polyphenols. Part 1: Insights of physico-chemical characterization
Authors: Lidija Fras Zemljič *, Olivija Plohl, Alenka Vesel, Thomas Luxbacher, Sanja Potrc
Received: 9 December 2019
Dear Editor, Dear Reviewers,
Please find below our response to the comments of both reviewers regarding our manuscript. We would like to thank the reviewers for their valuable comments, which helped us to improve the manuscript. All cited issues have been clarified to the best of our knowledge. We hope that the article is now suitable for publication in International Journal of Molecular Sciences -Submitted to section: Materials Science, Biopolymers as Food Packaging Materials.
A detailed list with explicit corrections and additions is presented below. All corresponding changes in the revised version of the manuscript are highlighted as track -changes.
Reviewer #2:
The title of the manuscript is too long, authors may short the title.
We thank the reviewer for these comments. However, we made a very wide research on the topic Packaging foils functionalized by colloidal synergistic formulation of polysaccharide chitosan and different polyphenols. We divided our work into two parts: i) physico-chemical characterization and ii) microbiological foils profile. In this way, two articles are prepared and both will be submitted in this special section. In order to separate both research stories, the title is longer, but somehow logical when research in is presented in two parts.
The abstract is not clear, authors need to put more effort to highlight their work more concisely. In abstract “Two representative, commercially used foils, polypropylene (PP) and polyethylene (PE), previously activated by UV/ozone, were functionalized using chitosan-extract nanoparticle dispersions”, as the functionalized particles are not in nano size, why author have used nanoparticles dispersions.
We thank the reviewer for these comments. The abstract was corrected in a suggested way (corrections are made by tracking changes in the article).
The procedure used for chitosan nanoparticles is ionic gelation that is known from the literature to provide nano-sized particles. The nanoparticles is meant for basic chitosan nanoparticles, which of course, were larger after encapsulation of polyphenols. Basic chitosan nanoparticles were of size: 293±80 nm. Because measurement provides hydrodynamic radio, pure particles (without water molecules surrounded) are smaller.
Moreover, nanoparticles are particles between 1 and 100 nanometres (nm) in any of size or dimension of particles. From SEM analyses it has been concluded that individually coated particles onto foils, also ranging from 100 nm to less than 1 μm.
Authors should mention the reasons why they have used UV/ozone? And explain the mechanism how it affects the surface of Polypropylene and Polyethylene.
We thank the reviewer for this comment; It might be that this was overlooked as it was already given in the original manuscript on the page 7:
As shown from previous work [12], for the untreated (pristine) PE and PP foils mostly carbon was detected, which is in agreement with the chemical composition of the virgin foils. Small amounts of oxygen (approx. 1 at. %) were present due to the surface contamination, as often observed in XPS spectra measured on pure polymer foils. After oxygen plasma treatment was used there, the oxygen concentration increased to 13.7 at. % and 14.7 at. % for PE and PP, respectively [12]. Similar, and even better, can be seen here by UV/O3 activation of foils, where the oxygen concentration increased to 19. 4 at. % and 15.1 at. % for PE and PP, respectively. Higher oxygen concentration on a PP sample can, again, be explained by oxidation of the side methyl group, which is absent on PE. Increased oxygen concentration indicates clearly the introduction of oxygen – functional groups that may act as a binding place for chitosan with embedded polyphenols. This is also proven by the presence of C-O, C=O and COOH groups, as seen from the high resolution XPS spectra shown in Figure 2.
Morphology of these foils is discussed in section SEM –page 14: For non-treated PE and PP foils a rough surface with some defects or impurities on the surface were seen, that may originate either from contaminated foils or impurity that may be introduced during the foil preparation for the analysis. When foils were treated with UV/O3 activation, quite smooth and clean surfaces were present on the images, especially in the case of PP-activated foil. It is known that UV/O3 clean the surface, which is reflected in foils` surface morphology in a smoother manner [44].
However, additional sentence was also added into abstract.
The size measurement performed with DLS, I am curious to know whether the size of CSNP THY 1138±93 , CSNP ROS 6627±353 and CSNP CIN 2993±435 DLS is good for below 1000nm, author have measure size 6627 with error 353nm and PDI is very abrupt, please recheck the use of DLS instrument.
We thank the reviewer for these comments. This kind of standard deviation SD is normal for measurement of larger and more size-dispersed particles. Mostly, dispersions showed size heterogeneity of particles and thus wider size distribution that increase the SD. lf we take into account the SD of method, the trend of results is still the same. The result are trustable because for each sample 12 measurements were done in three parallels.
I am confused with sentence, “The positive zeta potential further confirms the accessibility of amino groups of chitosan, which suggests the encapsulation of the polyphenols into the chitosan nanoparticles”. As polyphenols are negative charge after coating with chitosan nanoparticles, it should reduce the charge of the particles.
We thank the reviewer for these comments. If zeta potential is positive this means that we detected positive amino groups as available surface groups on foils. Polyphenols are negative and if they encapsulate into inner part of chitosan nanoparticles they are not accessible. Positive charge means that we detected wall of chitosan nano-capsulas/particles and that some amount of chitosan is still available. However, due to the increase of negative zeta plateau level, some extracts are accessible at the external foil surface. It may be concluded that we have available at the same time some extract s well as available chitosan.
The overall results and discussion needs to be improved.
We thank the reviewer for this comment. One paragraph was deleted. All other texts stay the same whilst to our opinion quite logical and extended presentation and discussion of results was done.
Authors can improve conclusion, which can highlight major finding and their prospective.
We thank the reviewer for this comment. The conclusion was corrected in a suggested way (corrections are made by tracking changes in the article).
Authors have not used any International Journal of Molecular Sciences references, As you are interested for this journal, if possible insert some related recent references from International Journal of Molecular Sciences.
We thank the reviewer for these comments. The reference was added
(number 8: Palza, H. Antimicrobial polymers with metal nanoparticles. Int. J. Mol. Sci. 2015, 16, 2099-2116.).
With best regards
Lidija Fras Zemljič